# Rapid Detection of Cd^2+^ Ions in the Aqueous Medium Using a Highly Sensitive and Selective Turn-On Fluorescent Chemosensor

**DOI:** 10.3390/molecules28083635

**Published:** 2023-04-21

**Authors:** Maria Sadia, Jehangir Khan, Rizwan Khan, Abdul Waheed Kamran, Muhammad Zahoor, Riaz Ullah, Ahmed Bari, Essam A. Ali

**Affiliations:** 1Department of Chemistry, University of Malakand, Chakdara 18800, Pakistan; 2Department of Electrical Engineering, Kwangwoon University Seoul, Seoul 01897, Republic of Korea; 3Department of Biochemistry, University of Malakand, Chakdara 18800, Pakistan; 4Department of Pharmacognosy, College of Pharmacy, King Saud University, Riyadh 11495, Saudi Arabia; 5Department of Pharmaceutical Chemistry, College of Pharmacy, King Saud University, Riyadh 11495, Saudi Arabia

**Keywords:** chemosensor, curcumin, heavy metals, aqueous medium, high sensitivity

## Abstract

Herein, a novel optical chemosensor, (**CM1** = 2, 6-di((E)-benzylidene)-4-methylcyclohexan-1-one), was designed/synthesized and characterized by ^1^H-NMR and FT-IR spectroscopy. The experimental observations indicated that **CM1** is an efficient and selective chemosensor towards Cd^2+^, even in the presence of other metal ions, such as Mn^2+^, Cu^2+^, Co^2+,^ Ce^3+^, K^+^, Hg^2+^,^,^ and Zn^2+^ in the aqueous medium. The newly synthesized chemosensor, **CM1**, showed a significant change in the fluorescence emission spectrum upon coordination with Cd^2+^. The formation of the Cd^2+^ complex with **CM1** was confirmed from the fluorometric response. The 1:2 combination of Cd^2+^ with **CM1** was found optimum for the desired optical properties, which was confirmed through fluorescent titration, Job’s plot, and DFT calculation. Moreover, **CM1** showed high sensitivity towards Cd^2+^ with a very low detection limit (19.25 nM). Additionally, the **CM1** was recovered and recycled by the addition of EDTA solution that combines with Cd^2+^ ion and, hence, frees up the chemosensor.

## 1. Introduction

Cadmium (Cd^2+)^ is one of the most hazardous and carcinogenic heavy metals, and it is widely employed in various industrial applications, such as the fabrication of metal alloys, batteries, electroplating films, and nuclear reactor control rods [1]. The effect of Cd^2+^ pollution should not be underestimated [2]. As a heavy metal ion with a biological half-life of 20 to 30 years, Cd^2+^ accumulates in the human body through contaminated water, air, soil, or other sources, leading to a variety of disorders of the kidney, liver, heart, lung, or other organs. Even if the accumulated Cd^2+^ content in the body is very low, it can still result in several health problems, including potentially fatal diseases, such as diabetes, cancer, and chondropathy [3]. Currently, there are several efficient methods to detect Cd^2+^: atomic absorption spectrometry (AAS) [4], inductively coupled plasma mass spectrometry (ICP-MS) [5], atomic fluorescence spectrophotometry (AFS) [6], the electrochemical method [7,8], and the fluorescence probe method [9,10,11]. Compared to the fluorescent probe technique, AAS, AFS, and ICP-MS need complex and expensive instruments, complicated sample preparation, and the electrochemical method, which has the disadvantage of poor selectivity.

The main advantage of the fluorescent probe technique is its high sensitivity, visibility, and rapid response. In addition, simple operation, low cost, and a wide linear range for heavy metal ions are also clear advantages. Therefore, the development of highly selective colorimetric/fluorescent chemosensors for the detection of harmful metal ions from environmental samples has received considerable interest due to their high sensitivity, selectivity, quick detection time, and cost-effectiveness [12,13,14]. In recent years, a large number of fluorescent probes with diverse scaffolds, such as quinoline, rhodamine, pyrene, boron-dipyrromethene (BODIPY), and anthraquinone, etc. have been developed with sensitivity, selectivity, and real-time detection of Cd^2+^ [12,15,16,17,18,19,20,20]. However, only a few examples of chemosensors have been reported where a considerable fluorescence enhancement is observed upon Cd^2+^ ion binding in aqueous medium [21,22,23]. The major difficulty in detecting Cd^2+^ is due to the relatively similar binding characteristics of Cd^2+^ with Zn^2+^ and Hg^2+^ cations, as they are in the same periodic table group. It is challenging to develop a probe for Cd^2+^ that does not exhibit the interference of Zn^2+^ and Hg^2+^ cations. Although a large number of fluorescent sensors have already been reported with success to differentiate Cd^2+^ from Zn^2+^ and Hg^2+^ cations, the majority of them have low water solubility with a lack of sensitivity and selectivity [24,25,26,27,28]. Therefore, designing and developing fluorescent probes with high water solubility and selectivity is a great challenge for the researcher. Herein, we report the design and synthesis of a water-soluble and simple fluorescent turn-on chemosensor comprising a **CM1** fluorophore, selectively detecting Cd^2+^ over other relevant metal ions in aqueous samples. Due to possessing the best coordination sites, it was capable of forming stable complexes with Cd^2+^ ions. The **CM1** showed a fast response, high sensitivity, and excellent selectivity for Cd^2+^ ion detection in an aqueous medium. The sensing mechanism of **CM1** was based on competitive binding with Cd^2+^ ions among different metal ions. Ethylenediaminetetraacetic acid (EDTA) was used as a chelating agent in reversibility studies. 

## 2. Results and Discussion 

### 2.1. Spectroscopic Studies 

The UV-Vis absorption spectra were investigated, and maximum absorption wavelength of **CM1** appeared at 390 nm, which was attributed to the π–π* transition (Figure 1A). Similarly, fluorescence investigation of chemosensor **CM1** was also carried out. When the chemosensor was excited at 390 nm, it gave an emission spectrum of low intensity at 610 nm, which could be attributed to the delocalization of oxygen lone pair electrons towards the aromatic ring, thus causing the quenching of fluorescence via the photoinduced electron transfer (PET) phenomenon (Figure 1B). The IR spectrum of the chemosensor **CM1** is depicted in (Appendix A). The IR spectrum gave the characteristic carbonyl (C=O) peak at 1710 cm^−1^, while the appearance of peaks at 1628–1646 cm^−1^ and 3000–2960 cm^−1^, respectively, corresponded to a C=C aromatic and C-H stretching.

### 2.2. UV–Visible and Fluorescence Study

The selective chemosensing ability of **CM1** towards Cd^2+^ was investigated through a detailed optical study. The UV–Vis absorption spectra of the **CM1** and **CM1**-Cd^2+^ complex are shown in (Figure 2). The chemosensor **CM1** gave maximum absorbance at 390 nm due to π–π* transition. Upon the addition of Cd^2+^, enhancement was observed at the absorption intensity at 390 nm. Similarly, the **CM1** fluorescence spectrum and its response to various metal ions was monitored through its excitation at a maximum wavelength (390 nm) in the aqueous medium. The addition of **CM1** to metal ions, such as Cd^2+^, Mn^2+^, Cu^2+^, Co^2+,^ Ce^3+^, K^+^, Hg^2+^, and Zn^2+^, showed very low enhancement in the fluorescence intensity, but maximum enhancement was observed for Cd^2+^ (Figure 3). This enhancement in fluorescence intensity was probably due to large rigidity in conjugation and better coplanarity upon complexation with the Cd^2+^ ion. In the case of other metal ions, the same phenomenon could not take place, thus inhibiting a significant enhancement in fluorescence intensity [29,30,31]. Potassium (K) and cerium (Ce) also showed a low fluorescence enhancement, and it may be possible that factors beyond the MLCT and LMCT effects, such as the shape and size of the metal ion, and its ability to accommodate the ligand, could be contributing to this similarity.

**Figure 1 molecules-28-03635-f001:**
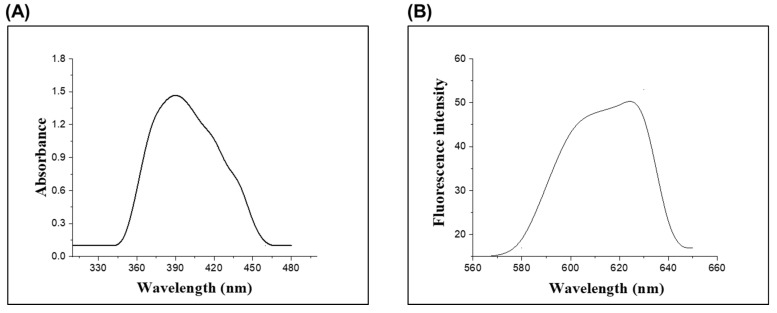
(**A**) UV-Visible absorption spectrum of chemosensor **CM1**. (**B**) Fluorescence emission spectrum of the chemosensor **CM1** when excited at 390 nm.

### 2.3. Stokes Shift

The reason for the observed large Stokes shifts was intra-molecular charge-transfer excitation of an electron from the HOMO to the LUMO of the chromophore, accompanied by elongation of the bond and considerable solvent reorganization due to hydrogen bonding to the solvent. The chemosensor **CM1** showed a prominent Stokes shift, and it can be used for practical application for the detection of Cd^2+^ [32].

### 2.4. Detection Limit and Quantum Yield

The LOD for the current chemosensor **CM1** was calculated to be 19.25 nM, as shown in Figure 4 [33]. The predictable mechanism of complex formation is given in Figure 1. 

The quantum yield calculated using Equation (2) for the current chemosensor **CM1** for Cd^2+^ detection was found to be 74%. Rhodamin 6G was chosen as the standard for quantum yield calculation because it is readily available and easily dissolvable in acetonitrile. Additionally, Rhodamone 6G has previously served as a standard for fluorescent chemosensors over a wide range of 340–540 [34].

### 2.5. Reuseability Study

The reuseability of the chemosensor **CM1** in the detection of Cd^2+^ ion is important for its practical applications. To check the reuseability of chemosensor **CM1**, a reversibility test was carried out using EDTA as chelating agent in aqueous medium at room temperature. The **CM1** can be freed and regenerated by the addition of EDTA to the Cd^2+^-**CM1** complex solution, which was observed through diminished fluorescence intensity. The enhanced fluorescence intensity was resumed by the addition of Cd^2+^ solution, which again combined with the **CM1** chemosensor. These results are a reflection of **CM1** as a reusable chemosensor in the presence of an EDTA solution (Figure 5). The results indicated that the same chemosensor, **CM1**, could be used for Cd^2+^ detection up to five times, with consistent and satisfactory outcomes.

**Figure 4 molecules-28-03635-f004:**
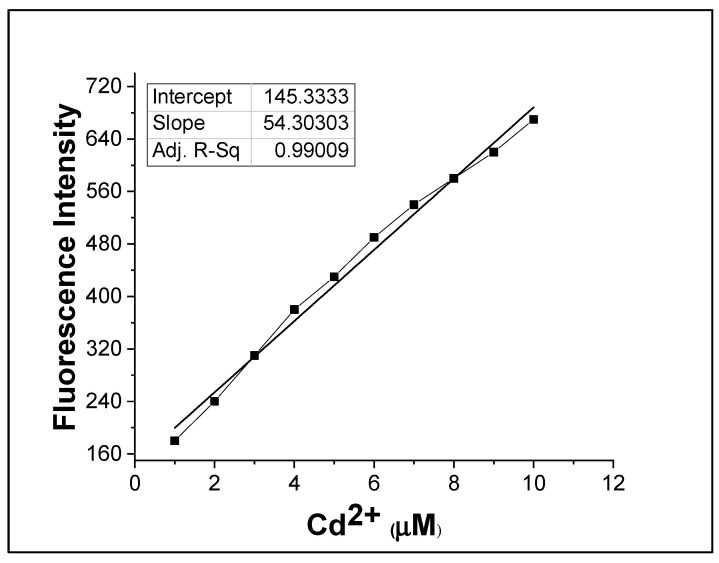
Calibration curve from fluorescence titration of Cd^2+^ (1–10 µM) range and **CM1** (10 µM) at λex = 390 nm and λem = 610 nm.

### 2.6. Effect of pH 

The dependence of pH variation on the fluorescence intensity of free **CM1** and its Cd^2+^ complex was investigated in the pH range of 2.0–12.0 (Figure 6). The pH of the test solution was adjusted with the help of a universal buffer. In case of a free chemosensor, fluorescence intensity remained constant from 2–12 pH, with weak fluorescence intensity. The low fluorescence intensity of **CM1** could be due to the photoinduced electron transfer (PET) phenomenon. However, for the Cd^2+^ (18 μM) complex with **CM1**, the fluorescence intensity gradually increased with pH. Comparatively low fluorescence intensity of the Cd^2+^ complex at lower pH was observed due to the protonation of carbonyl oxygen, which blocked the complexing ability of **CM1**. The maximum fluorescence intensity for the Cd^2+^ complex was obtained at pH 7 due to the restriction of PET. The fluorescence intensity decreased again with pH increase beyond 8 due to possible formation of sparingly soluble hydroxides complex. These results indicate that chemosensor **CM1** can be efficiently employed for Cd^2+^ detection at pH 7 [28].

The decreased fluorescence of the mixture of ligand and Cd^2+^ at both pH 3 and pH 12, as well as the lower fluorescence of the free ligand in the presence of excess cadmium, may be attributed to the formation of a non-fluorescent complex between the ligand and the metal ion. This complex may possess different structural and electronic properties compared to the free ligand, leading to a reduction in its fluorescence intensity. Additionally, factors, such as quenching, energy transfer, or changes in the solvent environment, may also play a role in the observed decrease in fluorescence. The decreased fluorescence of the mixture of ligand and Cd^2+^ at both pH 3 and pH 12, as well as the lower fluorescence of the free ligand in the presence of excess cadmium, may be attributed to the formation of a non-fluorescent complex between the ligand and the metal ion. This complex may possess different structural and electronic properties compared to the free ligand, leading to a reduction in its fluorescence intensity. Additionally, factors, such as quenching, energy transfer, or changes in the solvent environment, may also play a role in the observed decrease in fluorescence.

**Figure 6 molecules-28-03635-f006:**
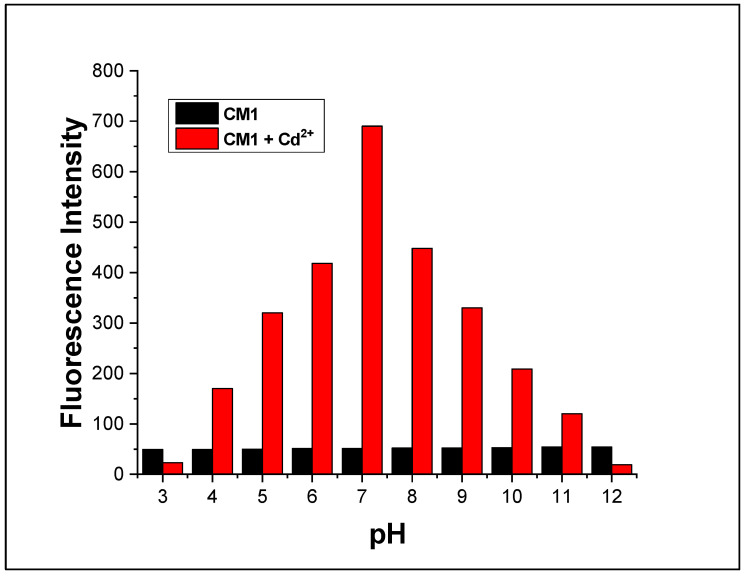
Effect of pH on fluorescence intensity of **CM1** and **CM1** (10 µM) with Cd^2+^ (18 µM) at λex = 390 nm and λem = 610 nm.

### 2.7. Anti-Interference Studies 

Anti-interference or competitive studies were carried out to verify the selectivity and specificity of the **CM1** for Cd^2+^ in the presence of other metal cations, such as Mn^2+^, Cu^2+^, Co^2+^, Ce^3+^, K^+^, Hg^2+^, and Zn^2+^. The results (Figure 7) showed that these interfering metal ions do not affect the fluorescence intensity of chemosensor **CM1**-Cd^2+^ complex even in the presence of higher concentration of interfering ions. Thus, this indicated that chemosensor **CM1** can be useful as a highly selective fluorescence-on sensor for trace determination of Cd^2+^ ions in different water samples [35,36,37].

### 2.8. Natural Water Samples Analysis by Spiking

Chemosensor **CM1** was applied for the selective detection of Cd^2+^ in lake water, tap water, and river water. Because, in these water samples, no Cd^2+^ was present, therefore a cadmium (2–10 μM) concentration range was added, and its fluorescence intensity was recorded at optimized parameters. From the calibration curve, respective Cd^2+^ concentrations were determined, and then % recovery was calculated. The results are shown in Table 1. Accordingly, these results demonstrated an excellent recovery of Cd^2+^ from these studied water samples. Thus, here we present a highly sensitive, selective, and cost-effective method for determination of Cd^2+^ in an aqueous medium with good % recovery [38]. 

### 2.9. DFT Studies 

The detection of Cd^2+^ with the help of chemosensor **CM1** was validated through computational investigation concerning the stability, orbital overlapping of Cd^2+^ and **CM1**, bond angles, bond lengths, and optimized geometry of the resulting cadmium complex. The optimized geometry of the **CM1** and its cadmium complex are shown in Figure 8. The **CM1** acted as a monodentate ligand attached through an oxygen atom to the Cd^2+^ ion, forming a stable distorted tetrahedral geometry.

The observed bond length, bond angles, metal–ligand interaction energy, charges on oxygen and cadmium, and highest occupied molecular orbital (HOMO) and lowest unoccupied molecular orbital (LUMO) energies of the cadmium complex are listed in Table 2. The observed Cd-O bond lengths with **CM1** were 2.24 Å and Cd-O bond lengths with a water value of 2.35 Å. The shorter Cd-O bond lengths with **CM1***,* as compared to water oxygen, was due to the extensive delocalization of electrons on chemosensor **CM1** aromatic rings that made their donor oxygen atom more electron-rich and, hence, caused strong interaction with an electron-deficient metal center. The observed O_CM-1_Cd-O_CM-1_ and O_CM-1_-Cd-O_water_ bond angles were 103.08 and 99.24°, respectively, suggesting a distorted tetrahedral complex. The tetrahedral geometry of the complex could also be explained based on orbital hybridization and their involvement in bond formation, since, in Cd^2+^, all the d orbitals are filled (d^10^ system). Therefore, no orbital in the “d” subshell was available for overlapping with chemosensor **CM1** filled orbital. Hence, the outer s and p subshells underwent sp^3^ hybridization and, consequently, formed tetrahedral structures. However, the small deviation from the regular tetrahedral structure was due to the bulky nature of the chemosensor **CM1**. The stability of the resulting Cd^2+^ complex was computationally calculated to be −14.35 eV, and the negative sign indicated the thermodynamic stability of the complex. It is worth noting that MLCT and LMCT effects, while useful for explaining metal–ligand interactions, are not the only contributing factors. Factors, such as the electronic configuration of the metal ion and the ligand, steric effects, and solvent effects, can also play a role in complexation. 

The interaction between Cd^2+^ and chemosensor **CM1** in the synthesized cadmium complex was also verified from the electron density difference (EDD) analysis. The EDD contour map shown in (Figure 9) indicated the concentration of a large electron cloud along the cadmium–chemosensor **CM1** axis. The HOMO and LUMO band gap gave information about the charge transfer interaction in a molecule. The electron transfer was efficient from the HOMO to LUMO, as the energy gap between these two orbitals became small. The negative value of HOMO and LUMO orbitals depicted in (Table 2) indicated the thermodynamic stability of the complex. The HOMO contained mostly the electron clouds on chemosensor **CM1**, while the Cd^2+^ ion contained a small electron density in LUMO, suggesting the transfer of an electron from chemosensor **CM1** to the cadmium ion upon complex formation. The smaller band gap (1.74) value suggested the utilization of the complex in semiconductor materials [39]. The formation of the cadmium complex was also supported by the extent of electron density transfer from both the oxygen atom of chemosensor **CM1** and water attached to Cd^+2^. As is obvious from Table 2, the formation of the complex caused a reduction in the electron density on the donor atoms of **CM1**, suggesting the formation of the complex. The absorption and fluorescence analysis showed that almost every metal atom was capable of forming a complex with the ligand. However, the highest enhancement was observed in the case of cadmium, indicating a high selectivity for this particular metal ion. Thus, we proceeded to investigate only the complexation reaction between the ligand and cadmium through DFT calculations. Our experimental results were supported by the DFT calculations, which confirmed the stability of the complex.

### 2.10. Comparison with REPORTED Chemosensors

The sensitivity of **CM1** was compared with previously reported chemosensors for the determination of Cd^2+^ ions. The result demonstrated that chemosensor **CM1** showed excellent sensitivity, as compared to the reported chemosensors (Table 3). The literature study also showed the lower limit of detection for chemosensor **CM1**, as compared to most of the recently reported similar work, indicating its superiority in terms of selectivity and practical applicability.

## 3. Materials and Methods

The chemicals and solvents used in the present study were analytical grade and utilized without further purification; the water employed in the experiment was double distilled. The metal salts of NiSO_4·_6H_2_O, Ca(OH)_2_·4H_2_O, Mn(OH_3_)_2_·4H_2_O, ZnSO_4_·6H_2_O, Cd(SO_4_)_2_·3H_2_O, Cu(SO_4_)_2_·5H_2_O), Co(OH_3_)·6H_2_O, CrCl_3_6H_2_O, KNO_3_, Hg(NO_3_)_2,_ Pb(NO_3_)_2,_ and Ce(NO_3_)_3_·6H_2_O were purchased from BDH chemical, England. As Ca(OH)_2_ was easily available in our laboratory, we utilized it as a source of calcium. Similarly, nitrate and sulphate salts of other metals were used. Sodium hydroxide, 4-methyl cyclohexanone, benzaldehyde, methanol, and acetonitrile were purchased from Sigma Aldrich and Merck. The UV-Visible spectra were obtained using a double-beam UV/VIS spectrophotometer (Shimadzu, Japan) model 1601. The fluorescent measurement was obtained through spectrophotofluorometer model no RF5301 PC (Shimadzu, Japan), containing a Xenon lamp as a source of excitation. The infra-red spectra were obtained through FT-IR spectrophotometer model 1601 (Shimadzu, Japan). The ^1^H NMR spectra were obtained on Prestige 21 (Shimadzu, Japan) using a Bruker Advance 400 MHz spectrometer. The spectrum was obtained in CDCl_3_ solution using tetramethyl silane (TMS) as an internal standard. 

### 3.1. Synthesis of Chemosensor **CM1**

The chemosensor **CM1***/*[(2,6-di((E)-benzylidene)-4-methylcyclohexan-1-one)] was synthesized as shown in (Figure 2) [48,49,50]. Ethanolic solution of benzaldehyde (1.06 g, 10 mmole) and 4-methyl cyclohexanone (0.49 g, 5 mmole) were mixed with continuous addition of sodium hydroxide (40%). The resulting mixture was refluxed for 4 h. The mixture was cooled to room temperature and evaporated to afford the desired product, **CM1***,* as a yellow-colored solid. The crude products were purified by recrystallization from ethanol. Melting point (M. P): 159 °C. Yield: 78%. IR (KBr, cm^−1^): 1710 cm^−1^ (C=O stretching), 1628–1646 cm^−1^ (C=C stretching), 3000–2960 cm^−1^ (C-H stretching) [51], ^1^H-NMR (300 MHz, ppm): δ 7.82 (bs, 2H, CH=C_6_H_5_), 7.34–7.50 (m, 10H, ArH), 3.07 (dd, *J_AB_* = 15.9, *J* = 3.6 Hz, 2H, CH_2_), 2.53 (m, 2H, 2CH_2_), 1.89 (m, 1H, CH), 1.08 (d, 3H *J* = 6.6 Hz, 3H, CHCH_3_) [52].

### 3.2. Solutions Preparation for Spectroscopic Measurements

Stock solution of chemosensor **CM1** in acetonitrile, and metal solutions using respective salts were prepared in distilled water. Stock solutions of metal (0.1 mM) were prepared in 100 mL solvent/water. Then working solutions were prepared by taking 5 mL from this stock metal solution and diluting it to 100 mL. Each time, a fresh working solutions of both chemosensor **CM1** and metal ion was prepared. 

### 3.3. General UV–Vis and Fluorescence Spectra Measurements

A stock solution of **CM1** (4 mM) was prepared in acetonitrile. Similarly, the same concentration solutions of the corresponding salts of copper, cerium, cadmium, zinc, mercury, cobalt, and manganese were prepared in double-distilled water. The tested solutions were prepared by mixing 50 μL stock solution of **CM1** with the appropriate amount of each metal ion in a tube. Absorption spectra were taken from the 250–500 nm range. Based on UV-Vis analysis, the fluorescent measurements were set at 390 nm. Each spectrum was taken three times, and their mean value was considered as final data. The chemosensing utility of **CM1** for Cd^2+^ in the presence of other metal ions was also investigated in natural water samples.

### 3.4. Excitation and Emission Spectra of Chemosensor **CM1**

Upon exciting chemosensor **CM1** at λex *=* 390 nm, its fluorescence emission spectrum was recorded in the range of 230–800 nm. A weak fluorescence emission spectrum was observed at λem *=* 610 nm. Thus, at these wavelengths of excitation and emission, further fluorometric analyses were carried out.

### 3.5. Limit of Detection and Quantum Yield Calculation 

The limit of detection (LOD) was calculated to be 19.25 nM from the calibration curve using Equation (1).
(1)$$\mathrm{LOD}=\frac{3\delta }{S}$$

In Equation (1) “𝛿” represents the standard deviation, and “*S*” is the slope of the fluorescence intensity vs. sample concentration curve.

Rhodamine 6G dye was used as a standard possessing a quantum yield (φ) 95 % for enumerating the quantum yield of the chemosensor **CM1** because Rhodamine 6G dye is readily available and easily dissolvable in acetonitrile. Additionally, Rhodamone 6G can be used as a standard for fluorescent chemosensors over a wide range of 340–540 [34,53]. For the experiment, Rhodamine 6G solution was prepared, and its spectroscopic analysis (UV-visible and fluorescence) were performed. The quantum yield was determined with the help of Equation (2).
(2)$$\varphi_{C\;=}\varphi_{R\;}\left(\frac{I_{C}}{I_{R}}\right)\left(\frac{\eta_{c}^{2}}{\eta_{R}^{2}}\right)\left(\frac{A_{R}}{\mathrm{Ac}}\right)$$
where “C” and “R” represent chemosensor **CM1** and dye, respectively. “I” represents the integrated fluorescence intensity, “η” corresponds to the refractive index of the solvent, and “A” corresponds to absorbance.

The values of absorbance in the equation were determined and used. For Equation (2), ${\;\varphi }_{R\;}$= 0.95, $I_{R}=620,\;\eta_{R}=1.33,\;A_{R\;=}$0.209, $I_{C}=40,\;\eta_{C}=1.34,\;\mathrm{and}\;A_{C}=1.5$.

### 3.6. Spiked Water Samples Analysis 

Natural water samples were collected from River Swat at Chakdara point, Malakand Lake at Dargai, and from the University of Malakand campus, respectively. Since no cadmium was found in these samples, the applicability of the new chemosensor **CM1** was determined by the spiking technique. A known amount of Cd^2+^ was added to each water sample, and the fluorescence intensity of spiked water samples was recorded. The selected water samples were spiked with different concentrations of Cd^2+^ (5–10) µM in the presence of chemosensor **CM1** (10 µM). The fluorescence response was measured at specific wavelengths of excitation and emission. For verification of the test results, each spectral analysis was carried out three times. 

### 3.7. Theoretical Calculation 

The density functional theory (DFT) calculations were employed utilizing DMol3 simulation code [54,55] The geometry and energy of the synthesized chemosensor **CM1** were optimized through Perdew Burke Ernzerhoffuncational (PBE) in the domain of generalized gradient approximation (GGA) [56,57]. The double numerical plus polarization (DNP) (Liu and Rodriguez 2005) was used for the adjustment of the basis set for all types of computation. Hessian calculations showed the absence of any imaginary frequency that reflects the stability of the relaxed structure. The thermal smearing parameters and basis set cut-off were adjusted, respectively, to 0.005 au and 4.6 Å. Grimme’s scheme DFT-D2 empirical dispersion correction was employed for van der Waals intermolecular interactions [58,59]. Convergence tolerance of 10^−5^ Ha for energy, 0.001 H/Å for force, and 0.005 Å for displacement were set during the geometries’ relaxation. The adsorption energies (E_int_) of complexes were evaluated using the following equation.
(3)$$E_{\mathrm{int}}{}_{}=E_{\mathrm{complex}}-(E_{\mathrm{CMI}}+E_{\mathrm{Cd}(\mathrm{II})})$$
where, E_complex_, E_lignd_, and E_Cd(II)_ are the total-electronic energies of the Cd(II)-**CM1** (complex), **CM1** and Cd(II) ion), respectively.

## 4. Conclusions

In summary, we have presented, here, the design and development of a novel optical chemosensor based on a curcumin derivative *CMI*, which can detect Cd^2+^ over other competitive metal ions in an aqueous medium. *CMI* exhibited good photostability and enhanced fluorescence in an aqueous medium upon the formation of the Cd^2+^ complex due to the presence of a carbonyl oxygen group. The interaction ratio between chemosensor *CMI* and Cd^2+^ ion was determined by fluorescence titration, Job’s plot, and computational studies, and it was found to be 2:1. The competitive binding experiment revealed no interference from the other metal ions (Mn^2+^, Ni^2+^, Cu^2+^, Co^2+,^ Ce^3+^, K^+^, Hg^2+,^ and Zn^2+^), thus showing selectivity of **CM1** towards Cd^2+^. Moreover, **CM1** showed high sensitivity at physiological pH, with low detection limits (19.25 nM). The **CM1***-*based chemosensor can be recycled and reused multiple times with the addition of an EDTA solution. This sensing approach certainly provides a basis that can be used for the design of various curcumin*-*based optical chemosensors to monitor other environmentally concerned hazardous heavy metal ions in aqueous medium.

## Data Availability

Not applicable.

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
