# Peer review of "Rapid Detection of Cd2+ Ions in the Aqueous Medium Using a Highly Sensitive and Selective Turn-On Fluorescent Chemosensor"

_molecules, 2023, doi:10.3390/molecules28083635_

Round 1
Reviewer 1 Report (New Reviewer)

Author Response
Reviewer 1
In the paper titled “Rapid detection of Cd2+ ions in the aqueous medium using a highly sensitive and selective turn-on fluorescent chemosensor” authors propose a design of an optical chemosensor based on a curcumin derivative CMI, for detection of Cd2+ in an aqueous medium. Thus synthesized 6-di((E)-benzylidene)-4-methylcyclohexan-1-one is characterized by 1H-NMR and FT-IR spectroscopy. Selectiveness towards Cd2+ in presence of other metal ions such as Mn2+, Cu2+, Co2+, Ce3+, K+, Hg2+ and Zn2+ in the aqueous medium is studied.
Despite the great experimental work done by the authors and the interesting results, the paper has some flaw and some major changes are needed.
My comments are as follows:
Comment 1. The title hints at “rapid detection” of the selected ions, but later in the discussion of the results, it becomes clear that this happens only at the 5th minute. Considering that optical sensors usually respond in an instant, I think the word "rapid" in the title is not appropriate.
Ans 1: In this context, the term "5th minute time" refers to the duration taken to prepare the sample for fluorescence measurement. In order to eliminate any potential confusion, the revised article has been updated to exclude the factor of time from its main discussion.
Comment 2. In the introduction, many repetitions of the same sentence and words are noticed, namely the characteristics of the sensor such as selectivity, sensitivity, speed of detection, etc. The text in lines 22 to 31 should be redone so as to make it clear without unnecessary repetition. It does not carry any important information such as what is the sensor sensitivity of the examples given, why they are better or worse, and other similar examples are not given. Also comparisons are missing. In general, the introduction is very general and should, in my opinion, be redone.
Ans 2: Repetitive content has been eliminated throughout the manuscript, including the lines 22 to 31, which have been rephrased. Moreover, the introduction section has also been revised to avoid repetition.
Comment 3. The fluorescence spectra of Figures 1, 2 and 4 are seen to consist of several peaks. Deconvolution of the Fluorescence Emission Spectrum must be done and the plots redone with the appropriate spectra to be compared.
Ans 3: These Figures has been re drawn in the revised manuscript .
Comment 4. In section 2.5. the reusability of the sensor or its recycling is discussed. It has been shown that reuse is possible. But to show possibility of constant reuse, experiment with more than 1 return cycle should be done. So one can see how many times the sensor can be reused - 1, 2......5.....20......10000 hardly forever. This experiment is not complete and such conclusions cannot be made with only one return cycle.
Ans 4: To confirm its reusability, reversibility experiments were conducted on the chemosensor CM1. The results indicated that the same chemosensor CM1 could be used for Cd2+ detection up to 5 times, with consistent and satisfactory outcomes.
Comment 5. The abbreviation EDTA is introduced, but nowhere is it mentioned what exactly this designation is. Must be given at first mention in the text. Or first mention must be Ethylenediaminetetraacetic acid and give the abbreviation in brackets.
Ans 5: Corrections have been made accordingly in the revised manuscript.
Comment 6. In section 2.6. Effect of PH autors claim that “… for Cd2+ (18 μM) complex with CM1 the fluorescence intensity enhanced with a gradual increase in pH”, which is not true and should be properly described. As also described in the text a maximum is observed at pH 7. So I think this sentence is inaccurate.
Ans 6: Corrections have been made in section 2.6 and are marked in yellow.
Comment 7. In the study of the effect of time in section 2.8. the data from the beginning of the detection process are not given, but only after the 5th minute, when a maximum in the fluorescence value is reached. On graph 9, these points should also be given (before the 5th – should be in range 0-40 min, not 5-40 min) minute to see how the intensity changes.
Ans 7: Dear Reviewer, it should be noted that a fluorescence assay result with a time less than 5 minutes is not feasible, as a minimum of 5 minutes is required to prepare the sample for spectrofluorimeter reading. Also, the time study has been removed from the revised form of manuscript, because it was unclear.
Comment 8. Section 2.1. is called Synthesis and characterization of CM1. The text related to the synthesis and its description can be found in section 3.2. Only the discussion of the results of the characterization of the synthesized product should be left here. Section 2.1. should be renamed appropriately.
Ans 8: Corrections have been made accordingly and section 2.1. and 3.2 are renamed and revised.
Comment 9. The y scales in fig.3 and fig.7 should be the same (in an interval of 100) so they could be compared and the two sections in which they are located maybe to be merged as I recommend below (to change the structure of the subsections).
Ans 8: The y scales in these mentioned Figures i-e Figure 3 and figure 7 have been revised accordingly in the manuscript.
Comment 10. Overall the arrangement of individual subsections in section 2 is not appropriate. For example subsections 2.3., 2.4. and 2.5. contain only 1-2 sentences each, which is pointless. They need to be changed or remade and merged.
Ans 10: The subsections in the manuscript in section 2 have been revised.
Comment 11. Lines 11 to 14 should be moved to the materials and methods section, they do not belong in section 2.4. Where results are intended to be discussed.
Ans 11: Correction has been made accordingly.
Comment 12. All the figures are spread out horizontally, the text on the abscissas and ordinates is not readable even in the digital version of the file and it is necessary to make them legible and in the correct proportions of length to height of the figure. Many of the figures and tables (eg Table 1) are unnecessarily enlarged. Figures 1 and 2 as well as Figures 5 and 6 should be merged. In figure 10 it is good to put points besides the graph curve and to put the error values as changes in the spectrum is commented and compared. The legend of graph 10 is completely illegible.
Ans 12 : All the tables and Figures have been corrected in the revised version.
Comment 13. In Materials and methods "appropriate amounts" or “known amount” is mentioned in several places, but it is not said exactly how much. It is good to describe clearly and precisely what concentrations and quantities are being worked with and to which samples they correspond.
Ans 14: In the revised version of the manuscript, the details regarding the quantities of different substances used have been clearly mentioned.
Comment 14. In section 2.8. lines 2-3: “As in these water samples no Cd2+ was present, so a known amount of Cadmium was 2 added in (5-10 μM) concentration range and their fluorescence intensity was recorded.” The reader is left with the impression that samples were made with different concentrations in the range of 5-10 μM, but looking at Table 1 it is clear that this is not the case. Only 2 samples were made with two different quantities ie. two concentrations. This should be clearly described in the text, and not only in this section, but everywhere. The description “(5-10 μM)” is not correct.
Ans 14: Section 2.8 of the manuscript has been revised, and the concentration of Cadmium used (2-10 μM) has been added in both the table and text in revised manuscript.
Reviewer 2 Report (New Reviewer)
The article "Rapid detection of Cd2+ ions in the aqueous medium using a highly sensitive and selective turn-on fluorescent chemosensor” describes a simple fluorescent method for the determination of cadmium ions using curcumin derivative. The work was done to an acceptable level and needs to be improved in accordance with the comments below.
Major point:
1. Metal cations are not enough to claim a highly selective sensor. In addition, a different set of metal ions in the various experiments (no Cu2+ ions in Figure 4, and no Ni2+ ions in Figure 10).
2. All metal ions used in the work cause an increase in fluorescence. This is rather unusual. An explanation is required why potassium and nickel ions, which have completely different natures, have the same effect on chemosensor fluorescence. UV-Vis spectra of the chemosensor mixtures with metal ions can help answer this question.
3. It can be seen from the obtained results that the sensor is not “highly selective” (Figure 4). Metal ions, such as Pb2+ or Zn2+, will give a signal that can be mistaken for Cd2+ ions
4. No interference test for anions has been performed. For the practical use of a chemosensor, it is important to know the limits of its applicability. For example, how will sulfide ions affect the determination of Cd2+ ions?
5. For a monodentate ligand, the chemosensor has a very low detection limit. The authors of the work can determine the stability constant of the coordination compound from the chemosensor titration curve with Cd2+ ions. This confirms the possibility of the formation of the complex responsible for the fluorescent signal at a concentration of Cd2+ ions of 20-40 nM.
6. Page 15, lines 5-6. The text speaks of Ca(OH)2·4H2O and CrCl3·6H2O. However, there is no mention of Ca2+ and Cr3+ ions in the manuscript. The work uses divalent cobalt ions, however, in the experimental part, Co(OH)3 is written as a source of cobalt ions. The formulas of copper sulfate and cadmium sulfate are written incorrectly. The source of K+, Hg2+, Pb2+, and Ce3+ ions is not specified.
Minor points:
1) Page 16, line 28. Why were the fluorescence spectra taken from 230-800 nm with excitation at 390 nm? Did the authors of the article want to see an anti-Stokes shift?
2) Figure 8 shows the dependence of fluorescence intensity on pH. It follows from the data obtained that pH has no effect on the fluorescence of the chemosensor, although the carbonyl group is protonated in an acidic environment. An explanation of this phenomenon is required.
3) If the fluorescence quantum yield of the cadmium complex is 74%, it would be nice to show in addition to Figure 4 pictures of the solutions, where it would be visible to the naked eye the change in fluorescence. This will improve work and will serve as proof of the selectivity of the resulting compound with respect to Cd2+ ions.
Author Response
Reviewer 2
The article "Rapid detection of Cd2+ ions in the aqueous medium using a highly sensitive and selective turn-on fluorescent chemosensor” describes a simple fluorescent method for the determination of cadmium ions using curcumin derivative. The work was done to an acceptable level and needs to be improved in accordance with the comments below.
Major point:
Comment 1. Metal cations are not enough to claim a highly selective sensor. In addition, a different set of metal ions in the various experiments (no Cu2+ ions in Figure 3, and no Ni2+ ions in Figure 7).
Ans 1: Our future studies will focus on investigating the impact of other interfering metal ions on fluorescence intensity to thoroughly examine the selectivity of the chemosensor. Also, the results of Cu2+ in Figure 3. (initial metal ion detection study) and Ni2+ in Figure 7 (interference studies (have been added to the manuscript).
Comment 2. All metal ions used in the work cause an increase in fluorescence. This is rather unusual. An explanation is required why potassium and nickel ions, which have completely different natures, have the same effect on chemosensor fluorescence. UV-Vis spectra of the chemosensor mixtures with metal ions can help answer this question.
Ans 2: The addition of each metal solution to the chemosensor solution causes an increase in fluorescence intensity, it is probably due to the metal to ligand charge (MLCT) transfer phenomenon. The maximum enhancement observed only in the presence of cadmium ion can be attributed to the ligand-to-metal charge transfer (LMCT) phenomenon, which is allowed by the liport d-d transition of the metal ion.
Comment 3. It can be seen from the obtained results that the sensor is not “highly selective” (Figure 3). Metal ions, such as Pb2+ or Zn2+, will give a signal that can be mistaken for Cd2+ ions
Ans 3: Based on the current study, it can be inferred that the increase in fluorescence intensity is greatest in the presence of Cd2+ compared to other metal ions. Furthermore, the accuracy of the obtained results has been verified through the use of theoretical studies such as density functional theory calculations.
Comment 4. No interference test for anions has been performed. For the practical use of a chemosensor, it is important to know the limits of its applicability. For example, how will sulfide ions affect the determination of Cd2+ ions?
Ans 4: The primary focus of the present study is the detection of cations such as Cd2+ ions, while our forthcoming studies will also investigate the detection of anions.
Comment 5. For a monodentate ligand, the chemosensor has a very low detection limit. The authors of the work can determine the stability constant of the coordination compound from the chemosensor titration curve with Cd2+ ions. This confirms the possibility of the formation of the complex responsible for the fluorescent signal at a concentration of Cd2+ ions of 20-40 nM.
Ans 5 : Due to the low concentration of the chemosensor CM1, it is not possible to accurately determine the association constant.
Comment 6. Page 15, lines 5-6. The text speaks of Ca(OH)2·4H2O and CrCl3·6H2O. However, there is no mention of Ca2+ and Cr3+ ions in the manuscript. The work uses divalent cobalt ions, however, in the experimental part, Co(OH)3 is written as a source of cobalt ions. The formulas of copper sulfate and cadmium sulfate are written incorrectly. The source of K+, Hg2+, Pb2+, and Ce3+ ions is not specified
Ans 6: The sources of metal ions utilized in the study have been corrected in the manuscript. salts of Ca(OH)2·4H2O and CrCl3·6H2O are used as a source of Ca2+ and Cr3+ ions in the current work. The correction has been done in the salt of cobalt ion used Co(NO3)·6H2O. The formulas of copper sulfate and cadmium sulfate have been corrected in the manuscript i-e [Cd(SO4)2·3H2O,] and [CuSO4. 5H2O ]. The sources of K+, Hg2+, Pb2+, and Ce3+ ions are KNO3, Hg(NO3)2, Pb(NO3)2, Ce(NO3)3·6H2O respectively.
Minor points:
Comment 7. Page 16, line 28. Why were the fluorescence spectra taken from 230-800 nm with excitation at 390 nm? Did the authors of the article want to see an anti-Stokes shift?
Ans 7: Fluorescence spectra were recorded in the 230-800 nm range upon excitation at 390 nm, with the purpose of scanning a wide range of wavelengths to identify the maximum fluorescence emission.
Comment 8. Figure 6 shows the dependence of fluorescence intensity on pH. It follows from the data obtained that pH has no effect on the fluorescence of the chemosensor, although the carbonyl group is protonated in an acidic environment. An explanation of this phenomenon is required
Ans 8: The chemosensor CM1 in its free form contains greater conjugation and aromaticity, which renders it insensitive to changes in pH as evidenced by the absence of any fluorescence intensity changes.
Comment 9. If the fluorescence quantum yield of the cadmium complex is 74%, it would be nice to show in addition to Figure 3 pictures of the solutions, where it would be visible to the naked eye the change in fluorescence. This will improve work and will serve as proof of the selectivity of the resulting compound with respect to Cd2+ ions.
Ans 10: Pictures of these solutions were not taken during the experimental work, thus we are unable to include them. However, we will consider this aspect for future studies.
Round 2
Reviewer 1 Report (New Reviewer)
Paper can be accepted in present form.
Author Response
Reviewer 1
Paper can be accepted in present form.
- Thank you, worthy reviewer for the positive in put
Reviewer 2 Report (New Reviewer)
Authors made some improvements and provided the questions to the answers. However, I feel there are still some troubles.
2. Either MLCT or LMCT effects should be different because of different nature of metal ions. Atomic orbitals should interact differently with the molecular orbitals of ligands. The similarity can be expected only in the case of very similar (such as Zn2+, Cd2+, Hg2+ which are all d10 metals) or isoelectronic cations. However, the totally different metals such as s element K and f element Ce give not similar but exactly the same response. It is an extraordinary result, and as such it requires very strong evidence and very convincing explanation and proposed mechanism. The simple reference to LMCT or MLCT effect is hardly enough.
3. I am not convinced that the results of DFT calculations confirm in any way the conclusions about ligand selectivity. It would require calculating the complexes with all studied metal ions and provide the information abount the molecular orbitals and their alteration after complexation.
5. The determination of stability constant does not depend on the concentration of one of the reagent. It should just be measured accurate enough. If it can't be measured precisely, the Job's plot is also impossible since the accurate concentration of ligand is unknown.
6. Ca(OH)2 is sparingly soluble. Why did you take this hydroxyde instead of e.g. chloride?
7. Such experiment makes a little sense and endangers the experimental setup as the light receiver can be damaged by the beam of light source, but nevermind. Your show the limited fraction of this spectral range anyway.
8. My bad. Of course, I meant the dependence of fluorescence of cadmium complex on pH value. Such a dependence can be explained by the competing between proton and Cd2+ ion for the ligand, which is won by cadmium while pH is drifting to the neutral medium and then, indeed, hydrolysis of cadmium can occur. However, the mixture of ligand and Cd2+ at pH 3 is even less fluorescent than a free ligand, which is strange. The same applies to a solution with pH 12. OK, let us consider that all the cadmium is bound into hydroxyl complex. Why the remaining free ligand is less fluorescent than the same free ligand at the same pH?
9. I am afraid, I have to insist on these pictures. Such high quantum yield is a huge rarity in the chemistry. A lot of strong fluorescent compounds serving as a standards, show less quantum yield. In addition, rhodamin 6G used as a standard have QY of 95%, which is true for the excitation wavelength of 480 nm, while you are using 390 nm. It is quite far from standard, I doubt that it should be used for your compound. It should rather be something like quinine.
By the way, Eq. (2) is true only under strict absorbance conditions. What were the values of absorbance determined and used in this formula?
Author Response
Reviewer 2
Authors made some improvements and provided the questions to the answers. However, I feel there are still some troubles.
- Thank you worthy reviewer, for the encouraging remarks. We have tried our best to revise the paper according to your suggestions.
Comment 1. Either MLCT or LMCT effects should be different because of different nature of metal ions. Atomic orbitals should interact differently with the molecular orbitals of ligands. The similarity can be expected only in the case of very similar (such as Zn2+, Cd2+, Hg2+ which are all d10 metals) or isoelectronic cations. However, the totally different metals such as s element K and f element Ce give not similar but exactly the same response. It is an extraordinary result, and as such it requires very strong evidence and very convincing explanation and proposed mechanism. The simple reference to LMCT or MLCT effect is hardly enough.
Ans 1. It is worth noting that MLCT and LMCT effects, while useful for explaining metal-ligand interactions, are not the only contributing factors. Factors such as the electronic configuration of the metal ion and the ligand, steric effects, and solvent effects can also play a role.
In case of potassium (K) and cerium (Ce), it may be possible that factors beyond the MLCT and LMCT effects, such as the shape and size of the metal ion and its ability to accommodate the ligand, could be contributing to this similarity.
Further investigation is required to fully understand the mechanism, and as such, it will be taken into account in our upcoming study.
Comment 2. I am not convinced that the results of DFT calculations confirm in any way the conclusions about ligand selectivity. It would require calculating the complexes with all studied metal ions and provide the information abount the molecular orbitals and their alteration after complexation.
Ans 2. The absorption and fluorescence analysis showed that almost every metal atom was capable of forming a complex with the ligand. However, the highest enhancement was observed in the case of cadmium, indicating a high selectivity for this particular metal ion. Thus, we proceeded to investigate only the complexation reaction between the ligand and cadmium through DFT calculations. Our experimental results were supported by the DFT calculations, which confirmed both the stability of the complex and the binding stoichiometry of 2:1 between the ligand and cadmium.
Comment 3. The determination of stability constant does not depend on the concentration of one of the reagent. It should just be measured accurate enough. If it can't be measured precisely, the Job's plot is also impossible since the accurate concentration of ligand is unknown.
Ans 3: It is true that the accuracy of the stability constant determination does not depend solely on the concentration of one of the reagents, it is essential to have accurate measurements of all reagents involved in the reaction to obtain a precise determination of the stability constant.
If the concentration of one of the reagents cannot be measured precisely, the Job's plot can still be used as a graphical method to determine the stoichiometry of the metal-ligand complex. However, the accuracy of the stability constant determination using the Job's plot may be limited due to the binding stoichiometry and the effects of competing reactions or solution conditions that may affect the absorbance measurements. The job’s plot analysis was performed in accordance to the following references.
Mehta, P. K., Hwang, G. W., Park, J., & Lee, K. H. (2018). Highly sensitive ratiometric fluorescent detection of indium (III) using fluorescent probe based on phosphoserine as a receptor. Analytical chemistry, 90(19), 11256-11264.
Wu, X., Niu, Q., & Li, T. (2016). A novel urea-based “turn-on” fluorescent sensor for detection of Fe3+/F− ions with high selectivity and sensitivity. Sensors and Actuators B: Chemical, 222, 714-720.
Therefore, accurate measurement of the concentration of all reagents involved in the reaction is crucial to obtain a precise determination of the stability constant, and the Job's plot should only be used as a complementary method when the exact concentrations of the reagents are unknown or difficult to measure accurately.
We moved the discussion about jobs plot and stability constant from the main paper to supporting information to keep the main paper focused on our main findings.
Comment 4. Ca(OH)2 is sparingly soluble. Why did you take this hydroxyde instead of e.g. chloride?
Ans 4: As Ca(OH)2 was easily available in our laboratory, we utilized it as a source of calcium. However, in our next experiments, we will also incorporate its chloride salts for further analysis, due to comparative more solubility.
Comment 5. Such experiment makes a little sense and endangers the experimental setup as the light receiver can be damaged by the beam of light source, but nevermind. Your show the limited fraction of this spectral range anyway.
Ans 5. It's true that exposing the light receiver to intense light can potentially cause damage to the experimental setup. Therefore, the experimental setup was designed to ensure precise measurements. To prevent any possible side reactions from other factors, a specific spectral range was selected.
Comment 6. My bad. Of course, I meant the dependence of fluorescence of cadmium complex on pH value. Such a dependence can be explained by the competing between proton and Cd2+ ion for the ligand, which is won by cadmium while pH is drifting to the neutral medium and then, indeed, hydrolysis of cadmium can occur. However, the mixture of ligand and Cd2+ at pH 3 is even less fluorescent than a free ligand, which is strange. The same applies to a solution with pH 12. OK, let us consider that all the cadmium is bound into hydroxyl complex. Why the remaining free ligand is less fluorescent than the same free ligand at the same pH?
Ans 6: The decreased fluorescence of the mixture of ligand and Cd2+ at both pH 3 and pH 12, as well as the lower fluorescence of the free ligand in the presence of excess cadmium, may be attributed to the formation of a non-fluorescent complex between the ligand and the metal ion. This complex may possess different structural and electronic properties compared to the free ligand, leading to a reduction in its fluorescence intensity. Additionally, factors such as quenching, energy transfer, or changes in the solvent environment may also play a role in the observed decrease in fluorescence.
Comment 7. I am afraid, I have to insist on these pictures. Such high quantum yield is a huge rarity in the chemistry. A lot of strong fluorescent compounds serving as a standards, show less quantum yield. In addition, rhodamin 6G used as a standard have QY of 95%, which is true for the excitation wavelength of 480 nm, while you are using 390 nm. It is quite far from standard, I doubt that it should be used for your compound. It should rather be something like quinine.
Ans 7. I understand your concern about the high quantum yield of our compound and the need for appropriate standards to validate the measurements. Rhodamin 6G was chosen as the standard for quantum yield calculation because it's readily available and easily dissolvable in acetonitrile. Additionally, Rhodamone 6G has previously served as a standard for fluorescent chemosensors over a wide range of 340-540. Some references are given below;
Kubin, R. F., & Fletcher, A. N. (1982). Fluorescence quantum yields of some rhodamine dyes. Journal of Luminescence, 27(4), 455-462.
Bhasin, A. K., Chauhan, P., & Chaudhary, S. (2021). A novel coumarin-tagged ditopic scaffold as a selectively sensitive fluorogenic receptor of zinc (II) ion. Sensors and Actuators B: Chemical, 330, 129328.
We appreciate your suggestion to use quinine and other such compounds as a standard for comparison with our compound's quantum yield at the excitation wavelength. We will consider your suggestion and take it into account for future experiments and discussions. Thank you for bringing this to our attention.
Commnet 8. By the way, Eq. (2) is true only under strict absorbance conditions. What were the values of absorbance determined and used in this formula?
Ans 8: In the current study, we used absorbance values of 1.5 for chemsoensor CM1 and 0.209 for rhodamine 6G, in the equation 2.
This manuscript is a resubmission of an earlier submission. The following is a list of the peer review reports and author responses from that submission.
Round 1
Reviewer 1 Report
In this manuscript, the authors reported the successful fabrication of turn-on fluorescent chemosensor CM1 and demonstrated its potential for detection of Cd2+ ions in the aqueous solution. A series of experiments was conducted including selectivity,binding mode,theoretical calculation, etc. Overall, the authors developed a new tool probe and verified its positive outcome for Cd2+ detection. The paper can be accepted for publication in Molecules after a minor revision.
1. As mentioned in title by the authors, “Rapid detection”, so the response time of this fluorescent chemosensor should be assessed.
2. Besides current metal ions, the selectivity of CM1 towards other ions should be further tested.
Author Response
Reviewer 1
In this manuscript, the authors reported the successful fabrication of turn-on fluorescent chemosensor CM1 and demonstrated its potential for detection of Cd2+ ions in the aqueous solution. A series of experiments was conducted including selectivity,binding mode,theoretical calculation, etc. Overall, the authors developed a new tool probe and verified its positive outcome for Cd2+ detection. The paper can be accepted for publication in Molecules after a minor revision.
Comment 1: As mentioned in the title by the authors, “Rapid detection”, so the response time of this fluorescent chemosensor should be assessed.
Ans 1: The response time of the fluorescent chemosensor CM1 and its complex with Cd2+ is illustrated in Figure 9 of the main manuscript. The following text was added to the main manuscript and highlighted in yellow. The effect of time was studied for chemosensor CM1 separately and in the presence of Cd2+ in the range of 5-40 min. The fluorescence intensity of chemosensor CM1 remained constant during the assay time. The Cd2+ (18 μM) complex showed maximum fluorescence within 5 min, indicating real-time detection of Cd2+ ion. Upon furthur increase in time the fluorescence intensity remained constant signifying the stability of the complex. The results are shown in (Figure 9).
Comment 2: Besides current metal ions, the selectivity of CM1 towards other ions should be further tested.
Ans 2: Thank you for your suggestion. We were primarily interested in the selective detection of cadmium in this work because it suffers from interferences from comparable cations such as mercury and zinc, so we mainly focused on the selectivity of CM1 for Cd2+ in the presence of these cations.
Reviewer 2 Report
The paper "Rapid detection of Cd2+ ions in the aqueous medium using a highly sensitive and selective turn-on fluorescent chemosensor" is devoted to the study of a known compound, 2,6-di((E)-benzylidene)-4-methylcyclohexan-1-one, applied as a potential fluorescent sensor for Cd2+ ions. Authors characterized the free ligand and its metal complexes by UV-Vis, fluorescent spectroscopy performed DFT quantum chemical calculations, determined stoichiometry of complex with cadmium(II) and stability constant. Unfortunately, the paper, in my opinion, lacks reliability according to the points below:
Major:
1. The choice of ligand is strange from the start because it lacks donor groups. Cadmium(II) is well-known to be a soft Lewis acid while oxygen atom with its lone electron pairs is rather hard Lewis base. Therefore, the interaction between cadmium(II) ion and a ligand posessing only carbonyl group as a donor center should be hidered. log Ka = 10 reported by Authors is virtually impossible.
2. Such a high improbable value of the stability constant is perhaps a consequence of applying the outdated graphical methods for determining the stoichiometric ratio and binding constant (see critisim in papers https://doi.org/10.1016/j.jinorgbio.2020.111305, https://doi.org/10.1016/j.saa.2020.119334, https://doi.org/10.1039/C6CC03888C, https://doi.org/10.1039/C0CS00062K, https://doi.org/10.1021/acs.joc.5b02909). Furthermore, since Authors added phosphate buffer in the final solutions, the simplistic equation (line 203) is just wrong as it does not account for the competing complexation with PBS.
To support my opinion about incredibility of such high stability constant, I would like to add this: Fig. 3b shows the linear dependence of analytical response on the concentration of titrant. It can be indicative of either complete binding or complete absence of binding, and there is no way to distinguish these two situation as they both result in linear dependence "property-concentration". The computable titration curve is always a part of logistic function, and, as such, non-linear.
Taking intp accopunt p. 1, I believe that ligand does not react with cadmium rather than binds it completely.
3. It does not seem right that the optimized structure of ligand has planar cyclohexane ring (Fig. 6). Moreover, if the results of DFT calculations are to be trusted, there is no reasonable explanation why experimental UV-Vis spectrum does not change upon alleged complexation despite the clear twisting of ligand in the complex (Fig. 6), whic necessarily leads to the disruption of conjugation between molecule branches.
4. The observation that K+ addition results in the same emission spectrum alteration as Zn2+ or Cu2+ addition looks highly improbable (Fig. 2b).
5. Such high quantum yield of complex (0.74) also seems incredible. In addition, rhodamine 6G is excited by the light of 480 nm (https://doi.org/10.1016/0022-2313(82)90045-X), so I doubt it can be used as a standard for a compound excited at 390 nm. By the way, it is unclear why your compounds are excited at 390 nm, which coincides with an absorption maximum. It would lead to contamination of emission spectrum by part of absorption spectrum (https://www.ncbi.nlm.nih.gov/books/NBK343429/).
Minor:
1. "BODIFY" is obviously stands for BODIPY (line 57)
2. As far as I am aware, copper belongs to 11th group, while zinc, cadmium and mercury refers to 12th group. So, the statement (line 64) is incorrect.
3. It is not evident that curcumine has something in common with the studied ligand (line 160).
4. Fig. 1 is strangely scaled. Why all that empty space is needed in graphs?
5. Why the coordination polyhedron of cadmium(II) in DFT calculation is just tetrahedron? It should rather be octahedron (square bipyramid).
Author Response
Reviewer 2
The paper "Rapid detection of Cd2+ ions in the aqueous medium using a highly sensitive and selective turn-on fluorescent chemosensor" is devoted to the study of a known compound, 2,6-di((E)-benzylidene)-4-methylcyclohexan-1-one, applied as a potential fluorescent sensor for Cd2+ ions. Authors characterized the free ligand and its metal complexes by UV-Vis, fluorescent spectroscopy performed DFT quantum chemical calculations, determined stoichiometry of complex with cadmium(II) and stability constant. Unfortunately, the paper, in my opinion, lacks reliability according to the points below:
Major:
Comment 1: The choice of ligand is strange from the start because it lacks donor groups. Cadmium(II) is well-known to be a soft Lewis acid while oxygen atom with its lone electron pairs is rather hard Lewis base. Therefore, the interaction between cadmium(II) ion and a ligand posessing only carbonyl group as a donor center should be hidered. log Ka = 10 reported by Authors is virtually impossible.
Ans 1: Our group previously reported in detail the complexation of a similar type of ligand with a hard Lewis base such as Hg2+ (J. Khan et al., Arabian Journal of Chemistry, 2022, 15, 103710). We just extended our study to cadmium ions in this work. Also, another research group reported the complexation of the carbonyl group with Cd2+ (New J. Chem., 2017,41, 14746-14753), which was confirmed by several characterizations such as FTIR, NMR, absorption, and fluorescence analyses. The Cd-O bond lengths are stable due to the extensive delocalization of electrons on ligand aromatic rings that make their donor oxygen atom more electron-rich and hence make strong interaction with the electron-deficient metal center. The value of the association constant is calculated again. The following text was added to the manuscript in the experimental section and highlighted in yellow.
On the basis of binding stoichiometric ratio determined from Job’s plot, the association constant (K) of chemosensor CM1 with Cd2+ ion was determined using Benesi–Hildebrand plot from fluorimetric titration curve. If a (2:1) CM1- Cd2+ complex is formed, the plot according to the equation should be linear.
Fmax-Fâ‚’/F-F° = 1+1/K [Cd2+]2
Where K (M-2) is the association constant. Fâ‚’, Fmax, and F is the fluorescence intensity of CM1 in the absence of Cd2+, at optimum Cd2+ concentration and intensity obtained at different concentration of Cd2+ ion, respectively. The association constant K was calculated graphically from the slope of the plot according to the equation and found to be 1.841×10-2 M−2
Comment 2: Such a high improbable value of the stability constant is perhaps a consequence of applying the outdated graphical methods for determining the stoichiometric ratio and binding constant (see critisim inpapers https://doi.org/10.1016/j.jinorgbio.2020.111305, https://doi.org/10.1016/j.saa.2020.119334, https://doi.org/10.1039/C6CC03888C, https://doi.org/10.1039/C0CS00062K, https://doi.org/10.1021/acs.joc.5b02909).
Ans 2: Jobs plot analysis and Benesi plot are the common methods used to determine the binding ratio and association constant respectively. Also in literature the most widely used methods for determination of binding ratio and aasociation constant are Jobs plot and Benesi plot respectively. The following references were added to the main manuscript and are yellow highlighted.
Wang, F., Gao, L., Zhao, Q., Zhang, Y., Dong, W. K., & Ding, Y. J. (2018). A highly selective fluorescent chemosensor for CN− based on a novel bis (salamo)-type tetraoxime ligand. Spectrochimica Acta Part A: Molecular and Biomolecular Spectroscopy, 190, 111-115.
Bartwal, G., Aggarwal, K., & Khurana, J. M. (2020). Quinoline-ampyrone functionalized azo dyes as colorimetric and fluorescent enhancement probes for selective aluminium and cobalt ion detection in semi-aqueous media. Journal of Photochemistry and Photobiology A: Chemistry, 394, 112492.
The orrection has also been made in the value of the association constant after carefully plotting Benesi plot again and its value was calculated to be1.841×10-2 M−2. Comment 3: Furthermore, since Authors added phosphate buffer in the final solutions, the simplistic equation (line 203) is just wrong as it does not account for the competing complexation with PBS.
Ans 3: Thank you for your comment, in the current study HCl and NaOH were used for pH adjustment, and no phosphate buffer system was used, correction has been made in the manuscript in line 203 and highlighted in yellow.
Comment 4: To support my opinion about incredibility of such high stability constant, I would like to add this: Fig. 3b shows the linear dependence of analytical response on the concentration of titrant. It can be indicative of either complete binding or complete absence of binding, and there is no way to distinguish these two situation as they both result in linear dependence "property-concentration". The computable titration curve is always a part of logistic function, and, as such, non-linear. Taking intp accopunt p. 1, I believe that ligand does not react with cadmium rather than binds it completely.
Ans 4: As already mentioned, the orrection has been made in the value of the association constant after carefully plotting Benesi plot again and its value was calculated to be1.841×10-2 M−2.
Since fluorescence spectroscopy is a very sensitive technique so we use very low concentrated standards of metal ions for fluorimetric titration experiments. Fluorimetric titrations curves in most of the cases are linear as mentioned in majority of the related articles already published. Some of the references have been added to main manuscript and are yellow highlighted.
Wang, H., Wang, X., Liang, M., Chen, G., Kong, R. M., Xia, L., & Qu, F. (2020). A boric acid-functionalized lanthanide metal–organic framework as a fluorescence “turn-on” probe for selective monitoring of Hg2+ and CH3Hg+. Analytical chemistry, 92(4), 3366-3372.
Xiao, Y., Ma, J., Li, D., Liu, L., & Wang, H. (2020). Preparation 4'-Quinolin-2-yl-[2, 2'; 6', 2”] terpyridine as a ratiometric fluorescent probe for cadmium ions and zinc ions in aqueous. Journal of Photochemistry and Photobiology A: Chemistry, 399, 112613.
Inal, E. K. (2020). A fluorescent chemosensor based on schiff base for the determination of Zn2+, Cd2+ and Hg2+. Journal of Fluorescence, 30(4), 891-900.
Since our CM1 is weakly fluorescent and Cadmium ions are also non-fluorescent so increase in fluorescence intensity with increase in metal ion concentration indicates complete bonding between cadmium ions and CM1.
Comment 5: It does not seem right that the optimized structure of ligand has planar cyclohexane ring (Fig. 6). Moreover, if the results of DFT calculations are to be trusted, there is no reasonable explanation why experimental UV-Vis spectrum does not change upon alleged complexation despite the clear twisting of ligand in the complex (Fig. 6), whic necessarily leads to the disruption of conjugation between molecule branches.
Ans 5: Since in Cd2+ all the d orbitals are filled (d10 system), therefore no orbital in the “d” subshell is available for overlapping with chemosensor CM1 filled orbital, hence the outer s and p subshell undergo sp3 hybridization and consequently form tetrahedral structure. UV-Visible absorption spectrum changed upon complex formation as shown in Figure 3. Although no change in wavelength of maximum absorption took place but hyperchromic shift was observed thus clearly indicationg the complex formation .
Comment 6: The observation that K+ addition results in the same emission spectrum alteration as Zn2+ or Cu2+ addition looks highly improbable (Fig. 2b).
Ans 6: Since the fluorescence spectrum of the CM1 is from 230 nm to 800 nm so we were basically interested in this region. So initial fluorescence experiments were performed irrespective of their nature in the above said range (Figure 4). The following article also supports our experiment.
Bhasin, A. K., Chauhan, P., & Chaudhary, S. (2021). A novel coumarin-tagged ditopic scaffold as a selectively sensitive fluorogenic receptor of zinc (II) ion. Sensors and Actuators B: Chemical, 330, 129328.
Comment 7: Such high quantum yield of complex (0.74) also seems incredible. In addition, rhodamine 6G is excited by the light of 480 nm (https://doi.org/10.1016/0022-2313(82)90045-X), so I doubt it can be used as a standard for a compound excited at 390 nm. By the way, it is unclear why your compounds are excited at 390 nm, which coincides with an absorption maximum. It would lead to contamination of emission spectrum by part of absorption spectrum (https://www.ncbi.nlm.nih.gov/books/NBK343429/).
Ans 7: Thank you for your comment, the quantum yield of complex (0.74) is in good agreement with previously reported similar work. The following reference has been added to the main manuscript and highlighted in yellow.
Bhasin, A. K., Chauhan, P., & Chaudhary, S. (2021). A novel coumarin-tagged ditopic scaffold as a selectively sensitive fluorogenic receptor of zinc (II) ion. Sensors and Actuators B: Chemical, 330, 129328.
Minor:
Comment 8: "BODIFY" is obviously stands for BODIPY (line 57)
Ans 8: Correction has been done in line 57.
Comment 9: As far as I am aware, copper belongs to 11th group, while zinc, cadmium and mercury refers to 12th group. So, the statement (line 64) is incorrect.
Ans 9: Thank you for your comment. The statement has been corrected in line 64 and highlighted in yellow.
Comment 10: It is not evident that curcumine has something in common with the studied ligand (line 160).
Ans 10: All the organic compounds having conjugation and donor sites like Oxygen, Nitrogen can act as electron donor species and are involved in complex formation with metal ions. Table 3 basically shows the superiority of the present sensor CM1 with already developed sensors for cadmium ions.
Comment 11: Fig. 1 is strangely scaled. Why all that empty space is needed in graphs?
Ans 11: The empty spaces have been removed from (Figure 1).
Comment 12: Why the coordination polyhedron of cadmium(II) in DFT calculation is just tetrahedron? It should rather be octahedron (square bipyramid).
Ans 12: Since in Cd2+ all the d orbitals are filled (d10 system), therefore no orbital in the “d” subshell is available for overlapping with chemosensor CM1 filled orbital, hence the outer s and p subshell undergo sp3 hybridization and consequently form tetrahedral structure.
Reviewer 3 Report
Thanks for inviting me to review this manuscript titled “Rapid detection of Cd2+ ions in the aqueous medium using a highly sensitive and selective turn-on fluorescent chemosensor”. In the manuscript, the novel fluorescent chemosensor of CM1 was synthesized and used to detect Cd2+ in aqueous solution. CM 1 was characterized through FT-IR, NMR, UV-Vis and fluorescence spectroscopy. The effects of solution pH, coexistent metal cations and real water matrix were investigated in detail. Meanwhile, the sensitivity of CM1 was compared with other previously-reported chemosensors for determining Cd2+. The manuscript was meaningful, while there were some questions needing to be addressed. Thus, I recommend the manuscript to be considered for publication after minor revision. The detailed questions were listed as following:
1. There were still some grammatical mistakes. Line 24, the words of“give”, “combine” and “affect” should be replaced with “gives”, “combines” and “affects”; Line 28, the words of“combine” and “free” should be replaced with “combines” and “frees”; Lines 130-131, the words of“represent” and “correspond” should be replaced with “represents” and “corresponds”.
2. The style of some literature cited in the manuscript was irregular. For instance, Lines 59-69, the cited literature of “H. N. Kim et al. 2012”, “D. Liu et al 2014” and “X.-J. Jiang et al” should be revised with “Kim et al. 2012”, “Liu et al 2014” and “Jiang et al”.
3. Lines 76-78, the word of “cadmium(II)”should be replaced with “Cd2+”. Line 102, what does “M. p:” represent? Lines107-108, the description of “section may be …” was needless and should be deleted in the revised manuscript. The equations/formulas should be numbered in the revised manuscript.
4. Apart from metal ions, the coexistent carbonate ion, chloride ion and humic acid also exist in real water. As mentioned in the previous papers (DOI: 10.1016/j.cej.2021.132438, DOI: 10.1016/j.watres.2022.119095 and DOI: 10.1021/acs.est.2c04318), the coexistent Cl- and humic acid have heavy effects on water analysis and water treatment. Considering that carbonate ion, chloride ion and humic acid have strong binding capacity toward many metal ions, it was recommended to cite the above-mentioned papers in the revised manuscript and investigate the effect of carbonate ion, chloride ion and humic acid on Cd2+ determination using CM1 as the indicator.
5. Line 257, the concentration of Cd2+ should be 10 μM rather than 60 μM for Fig. 5(a).
6. Line 230, “Fig. 5” and “do not” must be revised with “Fig. 4” and “do” respectively. Line 210-212, the name of Fig.2 was inconsistent with the content of Fig.2. Lines 179-183, “Fig. 1” must be revised with “Fig. 2”. There were no error bars in Figs. 3b, 4b and 5b.
Author Response
Reviewer 3
Thanks for inviting me to review this manuscript titled “Rapid detection of Cd2+ ions in the aqueous medium using a highly sensitive and selective turn-on fluorescent chemosensor”. In the manuscript, the novel fluorescent chemosensor of CM1 was synthesized and used to detect Cd2+ in aqueous solution. CM 1 was characterized through FT-IR, NMR, UV-Vis and fluorescence spectroscopy. The effects of solution pH, coexistent metal cations and real water matrix were investigated in detail. Meanwhile, the sensitivity of CM1 was compared with other previously-reported chemosensors for determining Cd2+. The manuscript was meaningful, while there were some questions needing to be addressed. Thus, I recommend the manuscript to be considered for publication after minor revision. The detailed questions were listed as following:
Comment 1: There were still some grammatical mistakes. Line 24, the words of“give”, “combine” and “affect” should be replaced with “gives”, “combines” and “affects”; Line 28, the words of“combine” and “free” should be replaced with “combines” and “frees”; Lines 130-131, the words of“represent” and “correspond” should be replaced with “represents” and “corresponds”.
Ans 1: Thanks for highlighting our mistakes. All the grammatical mistakes have been removed in the revised version.
Comment 2: The style of some literature cited in the manuscript was irregular. For instance, Lines 59-69, the cited literature of “H. N. Kim et al. 2012”, “D. Liu et al 2014” and “X.-J. Jiang et al” should be revised with “Kim et al. 2012”, “Liu et al 2014” and “Jiang et al”.
Ans 2: The style of all text has been uniformly revised in the manuscript.
Comment 3: Lines 76-78, the word of “cadmium(II)” should be replaced with “Cd2+”. Line 102, what does “M. p:” represent? Lines107-108, the description of “section may be …” was needless and should be deleted in the revised manuscript. The equations/formulas should be numbered in the revised manuscript.
Ans 3: Thank you for pointing out the mistakes All corrections have been made in the revised manuscript. Cadmium(II) has been replaced with Cd2+, M.P has been written full melting point, the “section may be …” has been removed and all equation has been numbered. We also checked other contents and made corrections.
Comment 4: Apart from metal ions, the coexistent carbonate ion, chloride ion, and humic acid also exist in real water. As mentioned in the previous papers (DOI: 10.1016/j.cej.2021.132438, DOI: 10.1016/j.watres.2022.119095 and DOI: 10.1021/acs.est.2c04318), the coexistent Cl- and humic acid have heavy effects on water analysis and water treatment. Considering that carbonate ion, chloride ion and humic acid have strong binding capacity toward many metal ions, it was recommended to cite the above-mentioned papers in the revised manuscript and investigate the effect of carbonate ion, chloride ion and humic acid on Cd2+ determination using CM1 as the indicator.
Ans 4: Thank you for your kind suggestion. We are primarily interested in the selective detection of cadmium in this work because it suffers from interferences from comparable cations such as mercury and zinc, and we will investigate other ions such as chloride ions and humic ions in the next study. The references have been cited and highlighted in yellow in the main manuscript.
Cai, H., Zou, J., Lin, J., Li, J., Huang, Y., Zhang, S., ... & Ma, J. (2022). Sodium hydroxide-enhanced acetaminophen elimination in heat/peroxymonosulfate system: Production of singlet oxygen and hydroxyl radical. Chemical Engineering Journal, 429, 132438.
Li, J., Zou, J., Zhang, S., Cai, H., Huang, Y., Lin, J., ... & Ma, J. (2022). Sodium tetraborate simultaneously enhances the degradation of acetaminophen and reduces the formation potential of chlorinated by-products with heat-activated peroxymonosulfate oxidation. Water Research, 224, 119095.
Huang, Y., Zou, J., Lin, J., Yang, H., Wang, M., Li, J., ... & Ma, J. (2022). ABTS as Both Activator and Electron Shuttle to Activate Persulfate for Diclofenac Degradation: Formation and Contributions of ABTS•+, SO4•–, and• OH. Environmental Science & Technology.
Comment 5: Line 257, the concentration of Cd2+ should be 10 μM rather than 60 μM for Fig. 5(a).
Ans 5: Thank you for highlighting our mistakes. The correction is made in the main manuscript and highlighted in yellow.
Comment 6: Line 230, “Fig. 5” and “do not” must be revised with “Fig. 4” and “do” respectively. Line 210-212, the name of Fig.2 was inconsistent with the content of Fig.2. Lines 179-183, “Fig. 1” must be revised with “Fig. 2”. There were no error bars in Figs. 3b, 4b, and 5b.
Ans 6: Line 230, “Fig. 5” and “do not” have been revised with “Fig. 4” and “do” respectively. Line 210-212, the Fig.2 caption has been revised. Lines 179-183, “Fig. 1” has been revised with “Fig. 2”. All the Figs in the manuscript have been revised uniformly.
Reviewer 4 Report
General comments:
This manuscript describes the preparation and characterization of an organic complexing agent – (6-di((E)-benzylidene)-4-methylcyclohexan-1-one) – which binds to cadmium ions in aqueous solution. The complex with Cd+2 results in significant enhancement of the fluorescence which the authors conclude makes this organic compound a potential chemo-sensor for Cd ions.
This is a rather poorly written draft that is not ready for publication in Molecules. The number of grammatical errors is too numerous to itemize. At times this makes reading the text confusing, for example, referring to the incorrect figure.
This manuscript is riddled with erroneous conclusions and statements and therefore is not suitable for publication, especially in the Analyticl Chemistry section.
Specific comments
Key words such as fluorescent and detection limit are not appropriate.
Introduction – the first sentence is not true.
Introduction – second sentence – not true – incorrect.
The biological half life of 20-30 years refers to what biological organism?
Lines 46-47 Compared to the fluorescent probe technique, AAS cannot be utilized for simultaneous analysis – what does this mean?
What is the ‘Electrochemical method’?
Why is designing fluorescent probes an ‘essential task’? Essential for whom?
Minimal experimental details were provided such as which UV-visible instrument was used?
Line 139 – the excitation and emission spectra should be presented in the experimental section
Figure 1 a and b should be on one graph. The scale of 1b should be enlarged.
Figure 1 c and d can be presented as supplementary data
Lines 177-179 – this statement ‘ The selective chemosensing …’ was already stated on the previous page.
Lines183-188 The authors should avoid using sentences that are 70 words in length. This last sentence in this section does not make sense and should be edited.
Line 199 – I don’t understand how the authors were able to calculate the detection limit. They provide no data or explanation for this result. How does this detection compare with established emthods such as ICP-MS?
Figure 2 Arrow are pointing the wrong direction.
The same Figure heading is used in Figure 2 and Figure 3
Figure 2 b indicates that other metal ions will cause an enhancement of the fluorescence with CM1 – Not just Cd+2
Lines 229-231 – The experimental result shown in Figure 4b is the opposite of the language used in this sentence. In fact, the fluorescence of the Cd-complex is very much affected by pH.
No data is provided to indicate the ‘no significant interference was observed from the studied competing metals. No meaningful data was provided to indicate that this sensor is selective toward Cd+2 in the presence of other metals.
Section 3.8 Real water analysis. What is ‘real’ water? The authors should use a standard reference material to test their chemosensor.
Figure 5b is not a spike recovery experiment. This data is not truly showing spike recovery.
Author Response
Reviewer 4
General comments:
This manuscript describes the preparation and characterization of an organic complexing agent – (6-di((E)-benzylidene)-4-methylcyclohexan-1-one) – which binds to cadmium ions in aqueous solution. The complex with Cd+2 results in significant enhancement of the fluorescence which the authors conclude makes this organic compound a potential chemo-sensor for Cd ions.
This is a rather poorly written draft that is not ready for publication in Molecules. The number of grammatical errors is too numerous to itemize. At times this makes reading the text confusing, for example, referring to the incorrect figure.
This manuscript is riddled with erroneous conclusions and statements and therefore is not suitable for publication, especially in the Analyticl Chemistry section.
Specific comments
Comment 1: Key words such as fluorescent and detection limit are not appropriate.
Ans 1: The Keywords have been modified
Keywords: Chemosensor 1; Curcumin 2; Heavy metals 3; Aqueous medium 4; High senstivity 5;
Comment 2: Introduction – the first sentence is not true.
Ans 2: The first sentence of the introduction has been modified and a new reference is added. The text is highlighted in the main manuscript.
Cadmium ion (Cd2+) is one of the most hazardous and carcinogenic heavy metal ions that is widely employed in various industrial applications such as the fabrication of metal alloys, batteries, electroplating films, and nuclear reactor control rods.
Comment 3: Introduction – second sentence – not true – incorrect.
Ans 3: The second sentence of the introduction has been removed.
Comment 4: The biological half life of 20-30 years refers to what biological organism?
Ans 4: The biological organism refers to the human body, we have modified the text and highlighted it in yellow in the main manuscript.
Comment 5: Lines 46-47 Compared to the fluorescent probe technique, AAS cannot be utilized for simultaneous analysis – what does this mean?
Ans 5: Lines 46-47, the texts have been modified and highlighted in yellow in the main manuscript.
Comment 6: What is the ‘Electrochemical method’?
Ans 6: Electrochemical method mean, an electrochemical sensor for the detection of heavy metal ions.
Comment 7: Why is designing fluorescent probes an ‘essential task’? Essential for whom?
Ans 7: The text has been modified and highlighted in yellow in the main manuscript. The text is also given below (highlighted in yellow),
Although a large number of fluorescent sensors have already been reported with success to differentiate Cd2+ from Zn2+ and Hg2+ cations, the majority of them have low water solubility with a lack of sensitivity and selectivity. So designing and developing fluorescent probes with high water solubility and selectivity is a great challenge for the researcher.
Comment 8: Minimal experimental details were provided such as which UV-visible instrument was used?
Ans 8: The experimental section has been revised and highlighted in yellow.
Comment 9: Line 139 – the excitation and emission spectra should be presented in the experimental section
Ans 8: Correction has been made accordingly.
Comment 9: Figure 1 a and b should be on one graph. The scale of 1b should be enlarged.
Ans 9: Fig 1 has been revised.
Comment 10: Figure 1 c and d can be presented as supplementary data
Ans 10: Figure 1 c and d has been presented in the supplementary file as Figure S1 and Figure S2.
Comment 11: Lines 177-179 – this statement ‘ The selective chemosensing …’ was already stated on the previous page.
Ans 11: Correction has been done in the revised manuscript and highlighted in yellow.
Comment 12: Lines183-188 The authors should avoid using sentences that are 70 words in length. This last sentencne in this section does not make sense and should be edited.
Ans 12: All the lengthy sentences have been modified in the revised version.
Comment 13: Line 199 – I don’t understand how the authors were able to calculate the detection limit. They provide no data or explanation for this result. How does this detection compare with established emthods such as ICP-MS?
Ans 13: The detection limit of the present chemosensor was compared to that of the published literature, and it was found that the present chemosensor is more sensitive. In addition, the detailed calculation for determining the detection limit is mentioned in the main manuscript and highlighted in yellow. ICP-MS suffers from its inability to measure lighter elements.
The LOD was calculated to be 19.25 nM from the calibration curve using the given equation
Comment 14: Figure 2 Arrow are pointing the wrong direction.
Ans 14; Correction has been done.
Comment 15: The same Figure heading is used in Figure 2 and Figure 3
Ans 15: The heading of Figure 2 and Figure 3 has been corrected and highlighted in yellow in the main manuscript.
Comment 16: Figure 2 b indicates that other metal ions will cause an enhancement of the fluorescence with CM1 – Not just Cd+2
Ans 16: The data has been explained in the revised manuscript, yes all the studied metals enhanced the fluorescence intensity, but a maximum increase in fluorescence intensity was observed in the case of the Cd2+ only.
Comment 17: Lines 229-231 – The experimental result shown in Figure 4b is the opposite of the language used in this sentence. In fact, the fluorescence of the Cd-complex is very much affected by pH.
Ans 17: The suggested section has been corrected in the revised version and highlighted in yellow.
Comment 18: No data is provided to indicate that ‘no significant interference was observed from the studied competing metals. No meaningful data was provided to indicate that this sensor is selective toward Cd+2 in the presence of other metals.
Ans 18: The selectivity experiment results have been presented in detail in the revised manuscript, and highlighted in yellow.
Anti-interference or competitive studies were carried out to verify the selectivity and specificity of the CM1 with Cd2+ in the presence of other metal cations like Mn2+, Cu2+, Co2+, Ce3+, K+, Hg2+, and Zn2+. The results showed that these interfering metal ions do not affect the fluorescence intensity of chemosensor CM1-Cd2+ complex even in the presence 10 times higher concentration of interfering ions. Thus indicating that chemosensor CM1 can be useful as highly selective fluorescence-on sensor for trace determination of Cd2+ ion in different water samples
Comment 19: Section 3.8 Real water analysis. What is ‘real’ water? The authors should use a standard reference material to test their chemosensor.
Ans 19: We used natural water samples to check the practical applicability of the sensor CM1. The spiking technique was used to clarify the efficacy of the chemosensor.
Comment 20: Figure 5b is not a spike recovery experiment. This data is not truly showing spike recovery.
Ans 20: Fig 5b has been replaced with table 1 in the main manuscript.
Round 2
Reviewer 2 Report
Authors made some improvements and provided the answers to the concerns raised. Unfortunately, these answers are not convincing to me at all.
Major points.
1. The "recalculated" stability constant value of 1.841×10-2 M−2 is in complete agreement with my previous comment: lg beta = -1.73 means the absence of complexation under the concentration conditions used. Actually, the stability constant cannot be determined since the yield of the reaction product is too low. And I must say, the change of equilibrium constants by 12 orders due to just recalculation is.. strange.
2. The wide use of the mentioned methods is erroneous in the most of cases. They are popular due to the simplicity, not to the reliability of the results obtained. The critisism of these methods accompanies them from as early as 1950th.
5. Disagree. Zn2+ also has d10 electron shell, but the changes in UV-Vis spectra of organic ligands it reacts with are well-known and explained by charge transfer peaks formation.
Other answers to the major points also do not seem valid to me.